# Reproductive Potential and Population Growth of the Worm *Enchytraeus buchholzi* (Clitellata: Enchytraeidae) Under Laboratory Conditions as Well as Regression Models

**DOI:** 10.3390/biology14020167

**Published:** 2025-02-06

**Authors:** Limin Zhao, Guilan Ma

**Affiliations:** 1School of Biological Science and Engineering, Shaanxi University of Technology, Hanzhong 723000, China; mgl2524@aliyun.com; 2Shaanxi Province Key Laboratory of Bioresources, Hanzhong 723000, China

**Keywords:** adult equivalent, cocoon growth, egg growth, *Enchytraeus buchholzi*, fecundity, polynomial models, population growth, reproductive potential

## Abstract

The worm *Enchytraeus buchholzi* is a key pest on American ginseng. Usually, it feeds on decaying vegetation, but it can become both parasitic and saprophytic when it meets with germinating seeds, living rootlets, and fresh taproots of the perennial, medicinal herb, causing significant injury. We conducted a study on individual and population reproductions, applied a series of bio-statistical methods to analyze the raw data, and constructed models based on their numerical increases. The results showed that the worm had a life span longer than 10-generation time, and revealed its daily reproductive potential and population growth trends. A parent adult laid 84.8 and 110.6 cocoons containing 545 and 714 eggs at 18 and 21 °C on average; the number of cocoons increased along a quadratic curve and that of eggs rose along a cubic curve. At 18–21 °C, it was able to reproduce 42 filial adult equivalents in a generation, and take place 7–9 generations a year in Liuba County, possessing a significant, generational reproductive potential. Its laboratory population followed a logistic growth pattern. The study has expanded our knowledge on bionomics and population ecology, and will help professionals predict population dynamics and promote ginseng production.

## 1. Introduction

American ginseng *Panax quinquefolium* is one of the medicinal crops in Liuba County, Shaanxi Province, China. However, diseases and insect pests infested the crop, resulting in significant damage [1,2]. In recent decades, some underground pests, including the worm *Enchytraeus buchholzi* Vejdovský (Clitellata: Enchytraeidae) [3,4,5] (Figure 1), were found to attack the perennial, medicinal herb by its roots, resulting in an abnormal reduction in seedlings in nursery beds and severe losses of yields [6] (Figure 2).

Specimens of the worm collected from Liuba County were sent to an institute for scientific identification, but they were misidentified as *Enchytraeus bulbosus* Nielsen and Christensen in 2014 [7]. Although the scientific name was incorrect, our project team has studied the worm in regard to the following aspects: pesticide screening [8], chemical control [9], physiological time [10], field temperature properties [11], bionomics [12], economic benefits and protective measures [6], and feeding amount and generational fertility [13]. The worm was re-identified into *E. buchholzi* in 2022, and thus recovered its original identity [4].

Geographic distribution of *E. buchholzi* is cosmopolitan; it is found in Great Britain, Ireland, Iceland, Norway, Sweden, Finland, Denmark, The Netherlands, Germany, Belgium, Portugal, Spain, Balearic Is., France, Switzerland, Austria, Italy, Greece, Poland, the Czech Rep., the Slovak Rep., Hungary, Romania; Tunisia, Algeria; Palestine, Rostov Province, Siberia, Primorskij Kraj, Turkey, Japan; Greenland, Canada; Brazil, Lake Titicaca, Argentina [14]. The list has been supplemented in recent years with the following regions: Bolivia, the Iberian Peninsula, Serbia, European waters (ERMS scope), the North Atlantic Ocean, Morocco, New Zealand [5], Colombia [15], South Africa [16], Russia (Kalmykia) [17], China (Liuba in Shaanxi, Fanjing Mountain in Guizhou) [4,18], and Norway (Spitsbergen in the High Arctic) [19].

*Enchytraeus buchholzi* is sometimes used as an experimental animal. It survived in cadmium (Cd)-polluted environments by aid of its Cd-inducible 25 kDa cysteine-rich protein (CRP); the data supported the view that the CRP gene was a unique MT-gene and has evolved by exon duplications from a MT-like ancestral gene [20]. Also, it was used to assess the ecotoxicity of gold nanoparticles (Au-NPs), Au(III), and their mixtures in bioassays; the order of toxicity was Au-NPs < Au(III) ≤ Au-NPs/Au(III) for reproduction [16]. In ecotoxicology, enchytraeids (including *E. buchholzi*) were used for the testing of chemicals mixed in artificial soil or of wastes and soils of unknown quality by applying a modified method, that is, extracting fixed and stained juveniles floating on the water surface, taking a photograph of the extract, and evaluating the digital image of the juveniles by computer processing [21]. The regeneration patterns were compared between a neoblast-bearing (*E. japonensis*) and a neoblast-lacking (*E. buchholzi*) samples, and the annelid neoblasts were suggested to be more essential for efficient asexual reproduction than for the regeneration of missing body parts [22]. *Enchytraeus buchholzi* accelerated the degradation of rice straw in a moist paddy soil; enchytraeids simultaneously reduced CO_2_ emissions by 35% on average, and their application was a promising management measure for stabilizing C in rice soils [17]. When both earthworms and enchytraeids (including *E. buchholzi*) were present, soil organic matter (SOM) mineralization did not increase; the effects of enchytraeids on SOM mineralization were similar with and without fungicide exposure [23].

Studying reproductive potential and population growth is an important work in animal ecology and is usually conducted by testing and analyzing the life tables of animals [24]. As a monoecious species, and about 5 mm long in adult stage, the worm feeds inside germinating seeds and roots of the host plant and has a population with overlapping generations. When field conditions were imitated and the worm was reared in wet sandy dishes in the laboratory, adults lived a very long life, laid many cocoons containing eggs in succession, and wrapped them with dust [13]. Individual worms with different ages were present in smaller wet sandy dishes and hid in food particles and sandy soils, which led to the situation that their number could not be accurately counted until the dishes were examined thoroughly by using a rinse method [13]. However, the rinse method was bound to change the rearing conditions, which would stop the conventional life table method from being applied in present study. Therefore, the authors turned to measure the individual fecundity of the worm in its lifetime, which led directly to a study of its individual reproduction. Meanwhile, the authors managed to estimate its laboratory population growth in another experiment through rearing the worm together with its offspring for a time longer than two generations.

Conventional theoretical ecology posits that an animal population experiences exponential growth under favorable conditions first and then logistic growth [24]. Here, we tested if the population of *E. buchholzi* follows the same growth trends even within its individual reproduction process.

We then reflected on its reproductive behavior by building statistical models and linking numerical growth trends together with population growth.

## 2. Materials and Methods

### 2.1. Preparation for the Worm

Roots of American ginseng damaged by underground pests were collected from Liuba County, Shaanxi Province, China. Living specimens of *E. buchholzi* were picked out from the roots and placed into plastic dishes containing both wet fine sandy soils and American ginseng powders to isolate and rear populations at a constant temperature of 21 °C. These populations served as our indoor stock colony for the following experiments.

### 2.2. Experimental Design and Execution

Applying a completely randomized design, the authors conducted two related experiments: (1) individual reproduction and (2) population reproduction. After that, we analyzed the data with statistical and modeling methods.

#### 2.2.1. Experiment I: Individual Reproduction—Measurement of Cocoons and Eggs Laid by a Parent Adult in Its Lifetime

Based on soil temperatures measured in the worm-infested region in early August [11] and the per capita reproduction and daily mean reproduction of the worm indoors [13], we selected 18 and 21 °C for the experimental treatments and conducted the experiment by rearing worms at the two temperatures: 14 adults at 18 °C and 14 adults at 21 °C, that is, 14 replications for each treatment. We used a total of 28 adults as parent and reared each of them alone in a 3.5 cm diameter wet sandy dish (see next paragraph for detail). Measurement of individual fecundity included: (1) the number of cocoons; (2) the number of eggs per cocoon; and (3) the number of eggs.

A batch of 28 polyethylene plastic dishes, 3.5 cm in inner diameter or 9.62 cm^2^ in inner area, was taken as rearing container. Both 4 g of sterilized fine sandy soils (passed through a 60-mesh screen; aperture = 0.30 mm) and 1.2 mL of distilled water were added to each dish, which was vibrated to flatten the wet sandy surface; extra water was sucked out, and then 10 mg of sterilized American ginseng powders (passed through a 50-mesh screen; aperture = 0.36 mm) was dusted onto the wet sandy surface of each dish. They were called the “wet sandy dish”, or “dish” for short, and used in the rearing experiment.

A total of 28 subadults [25,26] (a subadult is here an individual of the species that has entered into its preoviposition stage, when one or two gray-white eggs are formed in its oviduct but have not been laid out yet) were picked out from the indoor stock colony, and each of them was inoculated into a wet sandy dish alone. After inoculation, the 28 dishes were randomly divided into two groups, with each group transferred into a thermostatic cabinet (Type 250D, Jintan Jerier Electrical Appliance Co., Ltd., Changzhou, China) at 18 °C or 21 °C, respectively, and reared. The RH was about 100% above the wet sandy surface inside each dish. The two cabinets were placed in a shaded laboratory, with very weak sunlight diffused through the front glass window, and shifted naturally with day and night.

The worm inoculated into each dish was checked once every two days during the experimental or rearing time, when each dish was placed under a binocular stereo microscope, and the parent adult matured from initial subadult and its cocoon(s) laid inside the dish were found and recorded. A total of 98 records were obtained from each dish in the process. All the cocoons were removed or sampled from each dish when checked each time, and then combined to form a cocoon sample on that date [in regard to the variable “eggs per cocoon (*EPC*)”, a *t*-test showed that there was no significant difference (*p* > 0.05) between cocoon samples from both treatments on the same day, and thus they were combined together]. In the early stage of the microscopic examination, every cocoon in each sample was dissected immediately to record the number of eggs enveloped within; the frequency of dissection was changed to once every four days in the middle and later stages; as a result, a total of 1522 cocoons, or 52 cocoon samples combined, were dealt with in this way. After each checking, some sterilized American ginseng powders and water were added to each dish accordingly to maintain a proper living condition for the worm tested.

Experiment I was conducted under a semi-sterilized condition. In order to reduce food contamination caused by ambient miscellaneous microorganisms, a new batch of 28 wet sandy dishes was prepared in the same way every 30 to 40 days, and the parent adults were then moved from the old dishes to the new ones to live and reproduce till their cocoon output was extremely lowered. The experiment lasted for 196 days; the raw data before day 185 were adopted, but the rest were ignored because of many zeros.

#### 2.2.2. Experiment II: Population Reproduction—Measurement of Laboratory Population Growth During a Time Longer than Two Generations

Here, we set up different rearing time for adults as the experimental treatments, 8 in total, with each treatment repeated 7 times.

Another batch of 49 polyethylene plastic dishes was taken as rearing container. They were also prepared similarly to the wet sandy dishes shown in Experiment I; a subadult was inoculated into each of them and reared at 21 °C in a thermostatic cabinet. Each of the subadults matured alone and then reproduced together with its offspring in the same dish to form a laboratory population. The parent adult and its offspring in each dish were observed once every two days during the rearing time, and sufficient sterilized American ginseng powders and water were added to each dish accordingly to maintain an appropriate living condition, with the other conditions the same as those in Experiment I.

The 1st sample was a visual one and obtained one day before an F_1_ cocoon was laid, with its date code assigned as day 0. The day when the incidence of new cocoons reached 50% was monitored in the 49 wet sandy dishes, labeled as the start of the F_1_ generation, and coded as day 1 in the following statistical analysis. With each sample containing seven wet sandy dishes, the 2nd one was drawn on day 3 and the 3rd to 8th ones were taken once every four days after day 11. With the exception of the first sample, the rest samples were each taken randomly, and a thorough examination was performed immediately, that is, the contents in each dish were rinsed three times with 10 mL of water. The rinse solution was collected in a watch glass and examined under a binocular stereo microscope, with the sampling date and the number of adults, juveniles (=young worms) and solid cocoons in each dish recorded. Experiment II ended after 35 days, in which the subadults took 4 days to mature and then reproduced for 31 days together with their offspring. The period of 31 days was 3 days longer than the 2-generation time (GT, = length of immature and preoviposition stages) of the worm at the same temperature [10].

## 3. Statistical Analyses

### 3.1. Experiment I: Individual Reproduction

#### 3.1.1. Designation of Independent and Dependent Variables

Each sampling date was coded in an ordinal manner starting with day 0 when the subadults were inoculated, and the series of the codes was taken as an independent variable (*T*). The 0.5th, 2nd, 3rd, 4th, and 5th powers of each code in the *T* variable (*T*^1/2^, *T*^2^, *T*^3^, *T*^4^ and *T*^5^) were calculated, constituted several different series (datasets), and taken as more independent variables (*Ts*).

Based on 14 replicates from each treatment on each sampling date, corresponding mean survival rate (*SR*) of the parent adults was calculated, and so were the 2-day mean cocoons (2*DMC*) on the same day. Two variables were then calculated along the ordinal date codes based on 2*DMC*, as follows:

Daily mean cocoons (*DMC*) = each 2*DMC* was divided by 2;

Cumulative cocoons (*CC*) = each 2*DMC* was added one after another till the end.

The mean number of eggs per cocoon (*EPC*) was calculated based on each cocoon sample on that day. The *EPC* itself formed a variable and was used to calculate the other two variables:

Daily mean eggs (*DME*) = each *EPC* was multiplied by the *DMC* on the same day;

Cumulative eggs (*CE*) = each *EPC* was multiplied by the 2*DMC* on the same day to form each 2-day mean eggs (2*DME*), and the latter was added one after another till the end.

Except for the *SR*, variables *DMC*, *CC*, *EPC*, *DME,* and *CE* were each taken as a dependent variable (*Y*), respectively.

#### 3.1.2. Simulation of Relations Between *CC* or *CE* and *T* by Applying Exponential and Logistic Functions

Variables *CC* and *CE* were natural log-transformed accordingly. Via the least square method, two sets of linearized exponential and logistic equations were built through a correlation and regression analysis between the log-transformed values of *CC* or *CE* and *T* itself, and these linearized equations were then transformed back to real ones [24,27,28]. Meanwhile, against both the linearized and the real equations, the residuals between the observed and the fitted values (for the linearized equations, those between the transformed points and their estimates) were analyzed by using a chi-square goodness-of-fit test to find the probability if each of the newly built equations fit the observed (or transformed) points [29,30].

#### 3.1.3. Stepwise Regression Analysis and Residual Analysis for the Newly Built Polynomial Regression Models 

Through use of the matrix methods, each series of *Y* was regressed respectively on all the *Ts* in the backward stepwise regression procedure to build a polynomial regression equation. In each step, one of the *Ts* with the largest *p*-value (= with the smallest regression effect) was identified and eliminated on the basis, whether the *F*-test ratio for its partial regression coefficient was significant at *p* ≤ 0.01 [30]. After a linear (first-order), a quadratic (second-order), or a cubic (third-order) equation was constructed, a similar residual analysis was performed as described above [29,30]. Then, a derivative equation against each quadratic equation of *CC* or each cubic equation of *CE* was calculated, and the biological significances of the resultant intercepts (for the first term) and slopes (for the second and/or the third term) were specified and explained.

Both the advantages and disadvantages of the exponential, the logistic, the quadratic and the cubic equations for *CC* and *CE* were compared and determined according to the higher aptness given by the chi-square goodness-of-fit test via a residual analysis [30]. Figures illustrating the trends of points and theoretical curves were plotted against *T* in order to compare effects fitted by each of the functions.

#### 3.1.4. Division of the Filial Egg Stage into Many Substages and Conversion of the Worms in Each Substage into Generational Adult Equivalents (*GAE*)

Based on the GT of the worm (*m* ± *SD*), 18.3 ± 0.47 days at 18 °C and 14.0 ± 0.50 days at 21 °C [10], the filial egg stage ranging from days 3 to 185 was divided into 10 (at 18 °C) or 13 (at 21 °C) substages, with each substage equaling to a life cycle and containing a portion of *CE* or generational cumulative eggs (*GCE*). Taken as in a closed population without considering mortality, immigration, and emigration, each 2*DME* in the *GCE* was converted into adult equivalents (*AE*) [31] (*AE*, subadult(s) capable of reproducing soon) via being multiplied by the development rate of the eggs in the substage and summed to generational adult equivalents (*GAE*), which was then divided by the *GCE* to return an average development rate (*ADR*). The formula to calculate *GAE* was$$GAE=\sum_{i=1}^{m}E_{i}D_{i}/m$$
where *E_i_* = the number of eggs laid on the *i*th day in the substage, *D_i_* = duration of the eggs from the *i*th day to *m*, and *m* = the GT in days at the temperature.

### 3.2. Experiment II: Population Reproduction

#### 3.2.1. Transform of the Number of Mixed Worms in Each Wet Sandy Dish to *AE*

In the first sample, the number of parent adult in each dish was equal to an *AE* on day 0. In the second sample, the number of initial cocoon(s) in each dish was multiplied by 1/4 × 4 and then the resultant product was added directly to the number of parent adult to obtain the value of *AE* on that day because the mean number of eggs per cocoon laid in the first seven days was ≤ 4 and no juvenile appeared before then. Based on raw data recorded from the third to eighth samples, the number of mixed worms in each dish was transformed into *AE* according to the following formula:*AE* = No. adults × 1 + No. juveniles × 3/4 + No. solid cocoons × 1/4 × 8
where coefficients 1, 3/4 and 1/4 were the average development rate of the worm in its three life stages within the mixed population, respectively [10,13], and coefficient 8 was the mean number of eggs per cocoon in the same reproductive period.

The mean *AE* of the seven replicates in each of the eight samples was calculated.

#### 3.2.2. Designation of Variables and Their Correlation and Regression Analysis

The codes of the eight sampling dates, including days 0 to 31, were taken as the independent variable *T* and the eight means of *AE* from corresponding samples were designated as the dependent variable *Y*. Both exponential and logistic functions were applied respectively, and a linearized correlation and regression analysis of the natural log-transformed *Y* on *T* was conducted via the least square method; after back transformation, two real equations for the laboratory population growth were built. In the meantime, a chi-square goodness-of-fit test was applied to the two real equations [24,27,28,29,30]; when a higher fit probability was taken as the criterion, the proper regression model was determined and the trend of *AE* varying with sampling date codes was plotted.

The two experiments were conducted at the Shaanxi University of Technology, China, in 2015; their statistical analyses and theoretical studies were performed intermittently and completed in 2024.

## 4. Results

### 4.1. Experiment I: Individual Reproduction

#### 4.1.1. Survivorship of the Parent Adults

Throughout the rearing time, the *SR* of the parent adults was maintained at a higher level. At 18 °C, the *SR* was 100% through day 19 and then decreased gradually to 64.3% on day 185. At 21 °C, the *SR* was 100% through day 49 and then lowered gradually to 71.4% on day 185.

#### 4.1.2. Exponential and Logistic Equations for *CC* and *CE* as Well as Their Aptness

When simulated with an exponential or a logistic function, in the phase of linearized analysis, the correlation coefficients (*r*) of the log-transformed *CC* and *CE* (or *CC*’ and *CE*’ in Table 1) on *T* were all significant (*p* < 0.001) (Table 1). *χ*^2^ values of the linearized regression equations ranged from 5.48 to 37.1, indicating that their fit probabilities were close to 1.000 (Table 1). After the linearized regression equations were transformed back, the *χ*^2^ values of the resultant real equations increased sharply, ranging from 426 to 4469, meaning their fit probabilities were very close to 0, except those of the two real logistic equations for *CC* (*χ*^2^ = 71.0 and *p* ≈ 0.940 at 18 °C; *χ*^2^ = 98.5 and *p* ≈ 0.279 at 21 °C) (Table 1). For the resultant real equations, these linearized correlation coefficients were no longer valid.

#### 4.1.3. Polynomial Regression Models of Each *Y* on *Ts*

Stepwise regression analyses indicated that each of the five dependent variables, *DMC*, *CC*, *EPC*, *DME,* and *CE*, was closely correlated to *T* and/or *T*^1/2^, *T*^2^, *T*^3^ retained, with its linear or compound correlation coefficient, *r* or *R*, significant at *p* < 0.001 (Table 2). The *F*-test ratios for the partial regression coefficients of *Ts* retained in the linear, quadratic, or cubic equation were all significant at *p* < 0.001 (Table 2), meaning their regression effects were all strong. The chi-square goodness-of-fit test indicated each of the *χ*^2^ values was very low, meaning the probability that the estimates fit the observed points was larger than 0.9994; therefore, we concluded the null hypothesis *H*_0,_ that the regression models fit the linear, quadratic and cubic growth trends (Table 2).

#### 4.1.4. Daily Mean Cocoons, *DMC*

The number of *DMC* increased rapidly with *T*, reached its maximum between 7 and 9 days, and then fluctuated and decreased slowly; the trend was fitted with a linear function (Table 2; Figure 3). The intercepts of the two linear equations were positive, with 0.816 cocoon a day at 18 °C and 1.02 at 21 °C; the slopes were negative, ranging from −0.00374 to −0.00439, meaning their regression lines were trending downwards (Table 2; Figure 3). The regression line associated with 21 °C was located in the upper position, over the line with 18 °C, indicating that, in the range tested, the higher temperature was helpful for more cocoon outputs (Figure 3).

Points were found to fluctuate around the regression lines in Figure 3, largely due to the movement of parent adults into totally new wet sandy dishes every 30 to 40 days, which caused valleys and recoveries of *DMC*, appearing as several subcycles. They added some noises but did not affect the overall falling trends of *DMC*. Similar phenomena could also be perceived in other figures below.

#### 4.1.5. Cumulative Cocoons, *CC*

Cocoons appeared from zero to a certain number and rose day by day; *CC* increased steadily with *T* along a quadratic curve, with their midrange arched, reflecting the pattern for cocoon growth (Table 2; Figure 4). Living at 18 °C for 185 days, each parent adult laid 84.8 cocoons on average; however, at 21 °C the mean number was 110.6 (Figure 4).

The *χ*^2^ values of the *CC* quadratic models were only 4.82 and 10.7, meaning their fit probabilities approached 1.0000 (Table 2), whereas the fit probabilities of the real logistic equations of *CC* were 0.940 at 18 °C and 0.279 at 21 °C (Table 1). This comparison showed that the simulating effects of the *CC* quadratic models were better than those of the real logistic equations of *CC*. Figure 4 illustrated that the tracks of the real logistic equations of *CC* were systematically deviated from the observed points, inferior to those of the *CC* quadratic models. The real exponential curves of *CC* were more badly deviated from the points observed (Appendix A).

The derivative of each quadratic model for *CC* on *T* and *T*^2^ was a linear equation (Table 2), expressing the average efficiency of the parent adults’ laying cocoons. When broken down, the intercepts expressed the coefficient of daily reproductive potential, which was 0.801 cocoon a day at 18 °C and 0.923 at 21 °C on average (Table 2); their slopes represented the coefficient of reproductive resistance, which was −0.00377 and −0.00361, small but significant at *p* < 0.001 (Table 2).

#### 4.1.6. Eggs per Cocoon, *EPC*

Independent variables retained in the *EPC* cubic equation were *T*^1/2^, *T*, *T*^2^ and *T*^3^, and the *F*-test ratios for the partial regression coefficients were all significant (*p* < 0.001). The partial regression coefficient quantifying the variable *T*^1/2^ (*b*_1_ = 6.77) and its partial *F*-test ratio (*F_b_*_1_ = 30.6, *p* < 0.001) were each the largest, indicating this variable played the strongest role in constructing the *EPC* model (Table 2). In terms of absolute value, those partial regression coefficients quantifying *T*, *T*^2^, and *T*^3^ became smaller in order, and so did their partial *F*-test ratios, meaning their roles were lowered in the same way (Table 2).

Figure 5 illustrated that *EPC* varied with *T*^1/2^, *T*, *T*^2^ and *T*^3^ along a cubic curve. The estimate of *EPC* was 2.9 eggs on day 3 of the rearing time, increased to the peak 8.1 on day 24, decreased gradually to 5.6 on day 90, nearly leveled off to 5.4 on day 150, and then lowered slowly to 3.3 on day 185. The daily mean of *EPC* was apparently in constant change as the rearing time extended.

#### 4.1.7. Daily Mean Eggs, *DME*

Independent variables retained in the *DME* cubic equations were also *T*^1/2^, *T*, *T*^2^ and *T*^3^, and the *F*-test ratios of their partial regression coefficients were all significant (*p* < 0.001). They played almost the same role as seen during the building of the *EPC* model (Table 2).

The tracks of the *DME* cubic curves ascended sharply in their initial stage, reached peaks around day 20, descended gradually for about 70 days, leveled off for about 60 days, went down again, and terminated almost on day 185 (Figure 5). At 18 °C, the estimates on days 20, 70, 90, 150 and 185 were 6.5, 3.1, 2.3, 1.7 and 0.1 eggs, while, at 21 °C, the estimates were 8.4, 3.7, 2.9, 2.8 and 0 eggs (Figure 5). The trends of *DME* illustrated that about 60% of the filial eggs were laid (though by means of cocoons indirectly, the same below) in the first 70 days, which was as long as nearly 4- or 5-folds of a GT of the worm at the temperatures tested [10] (Figure 5).

#### 4.1.8. Cumulative Eggs, *CE*

Shown in Table 2 and Figure 6, the tracks of the *CE* increased steadily with *T*, *T*^2^ and *T*^3^ along a cubic curve, and the arch in the midrange of each curve was more strengthened. Living at 18 °C for 185 days, each parent adult laid 545 eggs on average; at 21 °C, the mean number was 714 (Figure 6).

The *χ*^2^ values of the *CE* cubic models were 34.9 and 48.8, meaning their fit probabilities were higher than 0.9999 (Table 2), whereas the fit probabilities of both the real exponential and real logistic equations for *CE* were close to 0 (Table 1). These comparisons indicated that the simulating effects of the *CE* cubic models were much better than those of the real exponential and real logistic equations for *CE*. Figure 6 illustrates that the tracks of the real logistic equations for *CE* were systematically and badly deviated from the observed points and, therefore, had to be discarded. The tracks of the real exponential equations (Appendix A) were completely departed from the observed points and thus also discarded. As a result, only the *CE* cubic models were preferred here.

The derivative equation of each cubic model for *CE* on *T*, *T*^2^, and *T*^3^ was a quadratic one (Table 2), expressing the average efficiency of the parent adults’ laying eggs, though the latter were enveloped in cocoons. When both derivative equations were broken down, the intercepts expressed the daily reproductive potential for a parent adult to lay eggs, which were 7.83 at 18 °C and 9.08 at 21 °C on average (Table 2); the coefficients for the second term *T* represented the reproductive resistance, ranging from −0.084 to −0.090 (Table 2). Those for the third term *T*^2^ were minute positives (0.000274 and 0.000293), apparently a little compensation to the daily reproductive potential and increasing with the square of the ordinal day.

#### 4.1.9. Values of *GCE*, *GAE*, and *ADR* as Well as Recognition of *R*_0_

Listed in Table 3, the values of *GCE* changed in different substages; the numbers of *GAE* that appeared in the first three substages were 42.5, 41.3 and 48.5 *AE* (with an average of 44.1 *AE*) at 18 °C and 41.2, 43.5, and 66.0 *AE* (with an average of 50.2 *AE*) at 21 °C, much more than those in other substages. Around means of 0.52, *ADR* varied from 0.41 to 0.62 at 18 °C and from 0.40 to 0.60 at 21 °C owing to the uneven numbers of partial *CE* in each substage (Table 3).

Now that *GAE* was given in a GT or a life cycle, the value 42.5 *AE* derived from the first substage could be recognized as the net reproductive rate *R*_0_ (or the net rate of increase per generation [24]) of the worm at 18 °C; in parallel, the value of 41.2 *AE*, as *R*_0_ at 21 °C. When the average of the *GAE* from the first three substages was chosen as an alternative, the mean values 44.1 and 50.2 *AE* could also be determined as *R*_0_ at 18 and 21 °C, respectively. The rest values of the *GAE* would be underestimates of *R*_0_ if they were used to do so (Table 3).

### 4.2. Experiment II: Population Reproduction

When fitted in an exponential function, *Y* (= *AE*, representing the laboratory population densities) was closely and positively correlated to *T* (*p* < 0.001) after the dependent variable was transformed into its natural logarithm (Table 4). The linearized regression equation played only an intermediate or transitive role (acted as a bridge) though the low *χ*^2^ value meant its fit probability was high (*p* > 0.9971). However, when the linearized equation was transformed back to a real exponential one, the *χ*^2^ value became very large, proving that its aptness was close to 0 (Table 4), which was not accorded with the reality and should be discarded. 

When fitted in a logistic function, *Y* was closely and negatively correlated to *T* (*p* < 0.001) after the dependent variable was transformed according to its special rule (Table 4). The *χ*^2^ value of the linearized equation was low, meaning its fit probability was very high (*p* ≈ 1.0000). When the linearized equation was transformed back to a real logistic one, the *χ*^2^ value was 6.519, indicating that its fit probability approached 0.4806 (Table 4), much larger than 0.05 and worthy of adoption (Table 4; Figure 7). Figure 7 illustrated that both the logistic and the exponential curves were close to each other on days 1–7 (day 7 was the midpoint of the first GT); afterwards, the logistic curve extended along the trend of the points observed, rose gradually, and then leveled off slowly, whereas the exponential curve was biased from the rest points, declining first, going lower and lower, turning back on day 23, rising up rapidly, and then badly deviating on day 31. The two curves crossed on day 26.4, near the end of the second GT, when the instantaneous population density was up to 471 *AE*/9.62 cm^2^, or 49 *AE*/cm^2^, acting as if it were a critical value. The difference between the two curves became larger and larger after that point.

## 5. Discussion

In Experiment I, the first part of present study, the authors started with measuring the fecundity of *E. buchholzi* in its lifetime. Either observed or theoretically fitted, the shapes of the fecundity curves reflected general trends in the individual reproduction of the worm. Because the worm lays eggs indirectly (it lays cocoons containing eggs), we conducted the measurement in three steps, which determined the specific procedure and content of Experiment I, distinct from those of Experiment II, which started with a population reproduction directly. It was worthwhile conducting both reproductive experiments indoors so as to find evidence to predict population growth trends in the field.

### 5.1. Polynomial Growth Trends in Individual Reproduction

As shown in Experiment I, higher survival rates, much longer life spans, and larger daily reproductive potential of the parent adults were detected. Compared with the GT in a broad sense, the filial egg stage, ranging from days 3 to 185, was as long as the period of time required for 10 or 13 full generations to take place at 18 or 21 °C, which proved that *E. buchholzi* belonged to a type of long-living species.

The derivative equations for the *CC* quadratic curves allocated coefficients for daily reproductive potential and resistance. Expressed by each intercept, the daily potential was the inherent power for parent adults to reproduce, while the resistance, represented by each slope, was their response to the increase in ages because the instant number or density of the worm in a dish was extremely low, or in other words, density-independent.

Although the intercepts and slopes given by the *CC* derivative equations were slightly different from those of the *DMC* linear models, the former provided adequate support for the latter; both achieved a mutual verification, showing that the daily reproductive potential interacted with the resistance each other, thus formed the actual number, the daily reproductive capacity, of *DMC* on each day. Compared with the *CC* quadratic models, the *CC* logistic equations showed a much lower aptness in regard to the points observed; therefore, they should be discarded.

Suitable simulations to the varying trends of *EPC* and *DME* were attained by applying cubic equations, including the square root of *T* as one of the independent variables. Both types of cubic curves reflected trends showing how the number of eggs changed with rearing time.

Based on continuous changes in *EPC* and *DME*, two *CE* cubic equations were built at 18 and 21 °C, which realized a sufficient simulation to actual facts. The derivative equations for the *CE* cubic models also differentiated the egg growth trend into coefficients of daily reproductive potential and resistance, even adding a tiny supplementary coefficient to the former, whereas, as indicated by the huge *χ*^2^ values, the real exponential and real logistic equations for *CE* exhibited extremely small aptness, showing that they lost contact with reality and had to be abandoned.

The common trends in the increment curves resulting from the *DMC*, *EPC* and *DME* equations were that points appeared initially in a low position, rose rapidly, fell down slowly, and vanished eventually, while those of the integral curves from the *CC* and *CE* equations were that their values increased more quickly in the early range than those in later one, and leveled off finally. The higher increments or more convex ranges in the early stage were induced by a vigorous reproductive capacity, whereas the lower increments or less convex ranges in the later stage were caused by the senescence of the parent adults. Just as shown above, resulting from individual reproduction, these trends could be described only by polynomial regression models but neither by exponential nor by logistic equation. Under this situation, as a portion of the whole reproductive process, the population in the first GT should have taken on a polynomial growth rather than others, even though it consisted of only filial eggs that would develop into adults in the future.

### 5.2. Basis and Application of R_0_

Both eggs and juveniles cannot be compared with adults in that their instantaneous reproductive capacities are zeros, which is totally different from the latter even though the former may grow up into the latter after completing a generational development. However, based on their development rates in a life cycle, a conversion of eggs and juveniles into *AE* made this comparison possible [13]. Following the same rule, both a reasonable division of the filial egg stage into many substages, with each equal to a GT, and a conversion of eggs in each substage into *GAE* were performed. Measured in the quantity of subadults, *GAE* represented a group of offspring reproduced by a parent adult in a GT, which was convenient to compare with results of previous studies on the one hand and revealed the important term *R*_0_ on the other hand.

Living together with offspring in a smaller wet sandy dish (1.5 cm in inner diameter) for a life cycle, the worm showed its actual generational fertility [13] (with the same meaning as current variable *GAE*). The values of *GAE* derived from the first substage of Experiment I (42.5 *AE* at 18 °C and 41.2 *AE* at 21 °C, as indicated above) were close to them and were, therefore, verified by an example from the population reproduction outside. Furthermore, estimated from the real logistic equation built in Experiment II, the laboratory population density was 43.5 *AE* (including 1 parent adult) in a wet sandy dish by the end of the first GT (day 14), approaching the *R*_0_ that was just obtained from Experiment I. Conducted under different rearing and sampling schemes, the three experiments had given estimates that were statistically similar to the value of *R*_0_.

In a broad sense, *R*_0_ is applied to estimate population growth of living things and calculate the economic threshold as well [32]. The exponential growth of a population is described by the formula *N_t_* = *N*_0_*e^rt^* [24], in which the power *e^r^* may be substituted by the term *R*_0_; if so, the formula became *N_t_* = *N*_0_*R*_0_*^t^*, an equivalent expression. If the following numbers were referred, for example, *N*_0_ = 1 (the initial number of parent adult), *R*_0_ = 41.2 (the number of *AE* by the end of the first GT), and *t* = 2 (the number of GT), the equivalent formula could estimate *N*_2_ = 1 × 41.2^2^ = 1697, meaning that the population size of the worm would reach 1697 *AE* by the end of the second GT (day 28) if the ambient resources, especially the rearing area, were unlimited.

### 5.3. Logistic Growth Trends in Population Reproduction

As shown in Experiment II, and as illustrated in Figure 7, the laboratory population density did not follow exponential but rather logistic growth when the worm was reared in the wet sandy dish for a period of time longer than two generations. Compared in terms of *AE* by the end of the second GT, the estimate given by the real logistic equation from Experiment II was only 533 *AE*, much lower than the estimate (1697 *AE*) from the hypothetical example given above. The difference between the two estimates was huge indeed, and the strong descending effect demonstrated the laboratory population was density-dependent. This enormous inhibition was actually caused by the over-crowding of more offspring born within the limited area inside the wet sandy dish. Referring to the information reflected in Experiment II, we may deduce that, the longer the duration that the worm is reared in a limited area, the stronger the inhibition that depresses its population density in the same site.

There arises a question herein: when to apply an exponential function and when a logistic one? According to the experimental study, a suggestion might be to apply an exponential function to simulate and describe a population size when it lives in an ecological niche with unlimited ambient resources or “under favorable conditions” [24]; otherwise, the first choice should be to apply a logistic function to simulate and estimate a population density if the ambient resources are limited and corresponding model is verified. This is similar to the example from Experiment II, in which one of the resources, the rearing area, was limited. Resources are often limited in the biosphere, and thus exponential growth is perhaps not as common as logistic growth in the field of population ecology if the period of observation is longer enough, lasting for 2, 3, or more GTs.

### 5.4. More Considerations

By the experimental study, we have found two kinds reproductive potentials, daily and generational; the former is expressed by intercepts of the derivative equations of polynomial growth models for cumulative cocoons and eggs, and the latter is indicated by adult equivalents generated by a parent worm in a full generation time, equaling to *R*_0_. Each has its own applying values.

An actual population possesses two attributes in an ecological niche, temporal and spatial, meaning that it exists and develops in a definite time and space. The laboratory population of *E. buchholzi* generated by population reproduction in Experiment II represents an actual population, whereas the laboratory population of *E. buchholzi* generated by individual reproduction in Experiment I had only temporal attribute; its spatial attribute was extremely weakened by artificially removing the cocoons on each checking day, leaving only a parent adult laying cocoon(s) in a dish. It was never a hypothetical population, as cocoon(s) appeared continuously. Once its spatial attribute were supplemented fully in its ecological niche, this kind of temporal population would put its generational reproductive potential into effect rapidly, leading to huge populations feeding on their host plant, American ginseng. Theoretically, the worm reproduces seven to nine generations a year [11], about 42 *AE* a generation, and an *AE* eats away 1.19 mg of fresh ginseng (excluding more tissues contaminated) [13]. This could potentially lead to devastating economic losses.

Modeling is one of the special research methods and needs to be verified by other theories and/or practices; theoretically, statistical tests can be effective to some extent. It should be noted that the chi-square goodness-of-fit test included in the residual analysis [30] is important, as it identifies if the fitted regression model is justifiable, especially when data transformation and a back-transformed equation are involved; the correlation coefficient can only be meaningful to the linearized form but invalid for the back-transformed equation reflecting the real relationship, as shown in present study. Only well-constructed real equations or models are able to make correct predictions and help solve practical problems.

It will be helpful to solve and explain some similar research problems if the methods used in the experimental study are applied to relevant fields.

## 6. Conclusions

On the basis of the results and discussions stated above, the authors reached the following conclusions:(1)The *DMC* linear equations and the *CC* quadratic equations were successful in fitting the trends of cocoon growth; the former described daily changes in cocoons and the latter expressed their accumulated effects.(2)The *EPC* cubic equation properly simulated changes in egg numbers in cocoons so that subsequent calculations and analyses would be continued.(3)The cubic equations of *DME* and *CE* were successful in fitting the trends of filial egg growth; the former displayed the daily increments of eggs and the latter reflected the reproductive outcomes of individual parent adults.(4)Polynomial regression equations were shown to be suitable models for individual reproduction of the worm *E. buchholzi* at 18 and 21 °C, whose daily reproductive potential and resistance were quantified by the derivative equations, and both interacted and formed its reproductive capacity, meaning that they acted as quantitative expressions of its reproductive behavior.(5)The *GAE* was calculated via a conversion of filial eggs into *AE*, and the *GAE* from the first substage functioned as the *R*_0_ of the worm. As a living material base accumulated by individual reproduction in F_1_ generation, and by taking *t* (the number of generations) as its index, the *R*_0_ would play an essential role in exponential growth of the *E. buchholzi* population size after the F_2_ generation if resources were unlimited.(6)Living together with its offspring in a limited area, the laboratory population of the worm bypassed an exponential growth and directly followed a logistic growth from the F_1_ to F_3_ generations.(7)The hypothesis of exponential growth was disproved by the population reproduction of *E. buchholzi* living in a limited area; neither exponential nor logistic growth was observed in the process and results of the individual reproduction of *E. buchholzi*, indicating that the conventional theory was not applicable in this aspect.(8)Revealing these statistical relationships helps professionals comprehend the individual reproduction of *E. buchholzi* clearly, understand the logical sequence and the difference between individual and population reproductions deeply, predict its population dynamics properly, develop its IPM program, and promote a stable and high productivity of American ginseng. The experimental study has expanded on theories pertaining to the bionomics and population ecology of *E. buchholzi* and opened a new area for research work in related fields.

## Figures and Tables

**Figure 1 biology-14-00167-f001:**
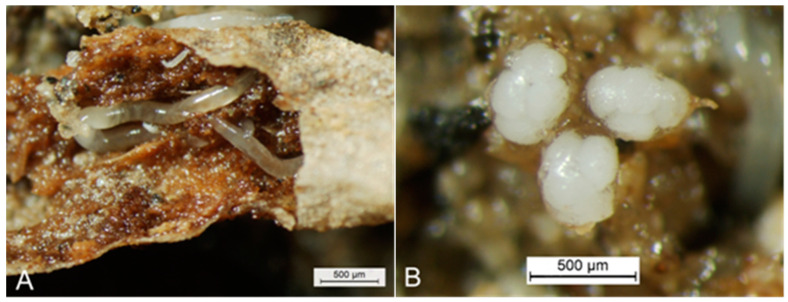
Adults of *Enchytraeus buchholzi* Vejdovský, 1878, with their egg cocoons [4]. (**A**) Three adults are injuring a one-year-old small root of American ginseng (the small root is peeled partially to show the interior). (**B**) Three cocoons of the worm, laid on the wet sandy surface by adults living inside the wet sandy dish, with each containing about five eggs.

**Figure 2 biology-14-00167-f002:**
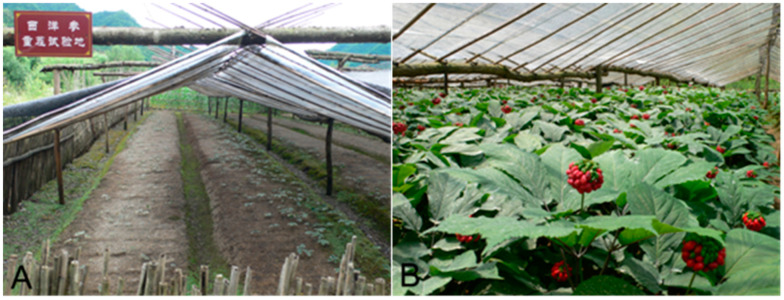
Plants of American ginseng grown within plastic greenhouses in Liuba County [6]. (**A**) Pesticide-free control plot in an experimental field for continuously growing American ginseng, showing most seedlings were dead and/or lost severely because of infestations by underground pests, including *Enchytraeus buchholzi* (Yingpan Village, July 2013). (**B**) American ginseng plants were healthy and strong, with luxuriant green leaves and red berries, predicting a bumper harvest of fresh roots; this is because efficient pest control and field management were executed regularly (Zaomulan Village, July 2013).

**Figure 3 biology-14-00167-f003:**
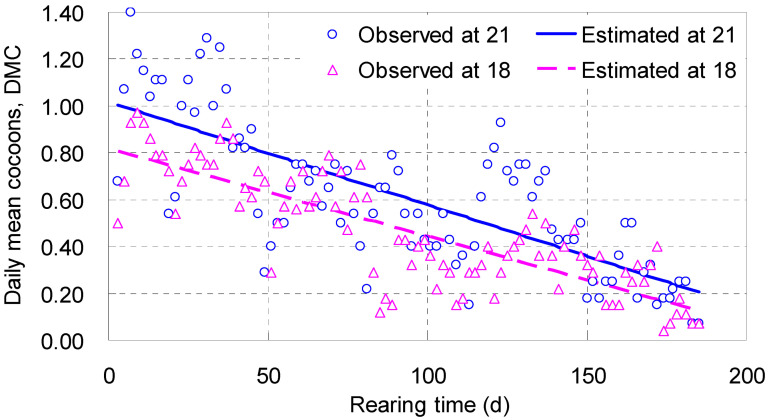
Increment curves of daily mean cocoons (*DMC*) laid by an adult of *Enchytraeus buchholzi* reared at 18 or 21 °C, respectively.

**Figure 4 biology-14-00167-f004:**
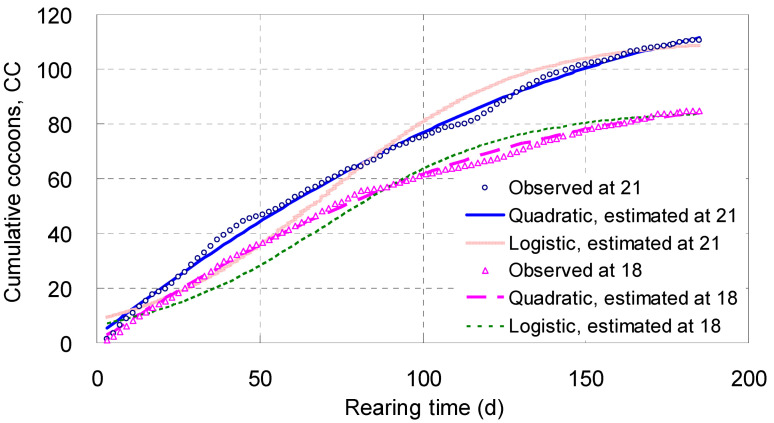
The integral curves of cumulative cocoons (*CC*) laid by an adult of *Enchytraeus buchholzi* reared at 18 or 21 °C, respectively.

**Figure 5 biology-14-00167-f005:**
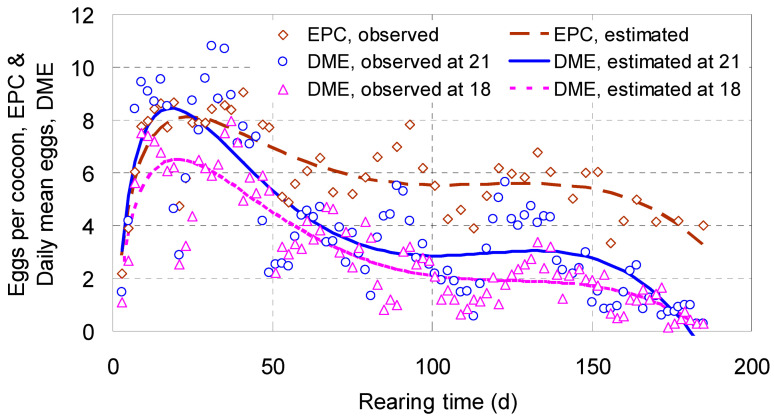
The increment curves of eggs per cocoon (*EPC*) and daily mean eggs (*DME*) laid by adults of *Enchytraeus buchholzi* reared at 18 and/or 21 °C.

**Figure 6 biology-14-00167-f006:**
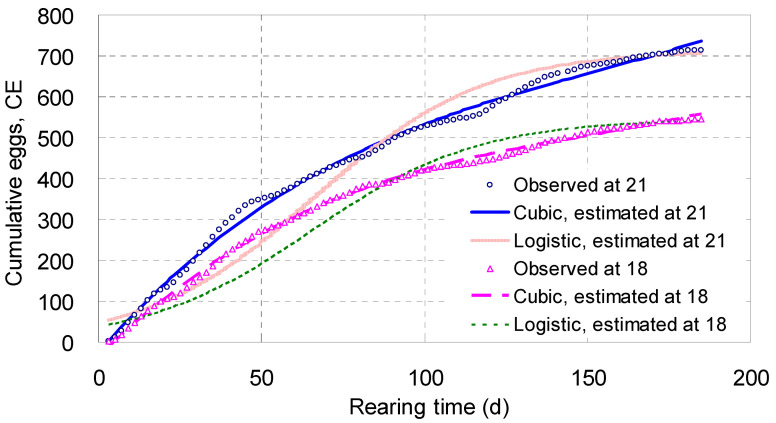
Integral curves of cumulative eggs (*CE*) laid by an adult of *Enchytraeus buchholzi* reared at 18 or 21 °C, respectively.

**Figure 7 biology-14-00167-f007:**
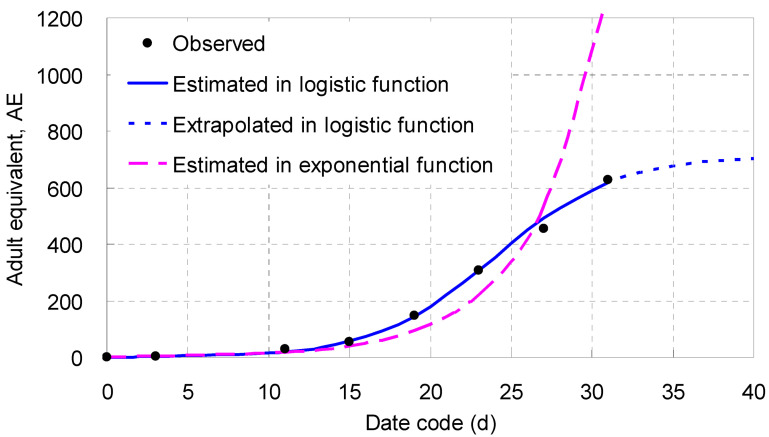
Growth trend of a laboratory population reproduced by an adult of *Enchytraeus buchholzi* reared at 21 °C for a period of time longer than two generations, expressed in adult equivalent, *AE* (its generation time = 14 days at 21 °C [10]).

**Table 1 biology-14-00167-t001:** Results of the simulation applying exponential and logistic functions, with cumulative cocoons (*CC*) and cumulative eggs (*CE*) acting as dependent variables and the rearing time (*T*) acting as an independent variable.

T. (°C)	Function	*n*	Linearized Phase		Chi-sq.-Test		Real Equation	Chi-Square-Test	
			*r*	Equation	*χ* ^2^	*p*≈		*χ* ^2^	*p*≈
18	Exponential	92	0.8192 ***	*CC*’ = 2.60 + 0.0125*t*	6.80	1.000	*CC* = 13.4 × 1.0126*^t^*	426 ***	9 × 10^−45^
			0.7477 ***	*CE*’ = 4.45 + 0.0129*t*	7.17	1.000	*CE* = 86.0 × 1.0130*^t^*	4002 ***	0.000
	Logistic	92	−0.9603 ***	*CC*’ = 2.48 − 0.0355*t*	20.7	1.000	*CC* = 85.0/(1 + 11.9*e*^−0.0355*t*^)	71.0	0.940
			−0.9449 ***	*CE*’ = 2.57 − 0.0392*t*	37.1	1.000	*CE* = 546/(1 + 13.0*e*^−0.0392*t*^)	745 ***	2 × 10^−103^
21	Exponential	92	0.8342 ***	*CC*’ = 2.86 + 0.0123*t*	5.48	1.000	*CC* = 17.5 × 1.0124*^t^*	441 ***	2 × 10^−47^
			0.7578 ***	*CE*’ = 4.73 + 0.0127*t*	6.27	1.000	*CE* = 113 × 1.0128*^t^*	4469 ***	0.000
	Logistic	92	−0.9657 ***	*CC*’ = 2.47 − 0.0346*t*	14.8	1.000	*CC* = 111/(1 + 11.8*e*^−0.0346*t*^)	98.5	0.279
			−0.9483 ***	*CE*’ = 2.59 − 0.0389*t*	32.6	1.000	*CE =* 714/(1 + 13.4*e*^−0.0389*t*^)	1100 ***	8 × 10^−173^

***, significant at *p* < 0.001. *CC*’ and *CE*’, natural log-transformed values of *CC* and *CE*.

**Table 2 biology-14-00167-t002:** Results of stepwise regression analyses for each of the five variables, daily mean cocoons (*DMC*), cumulative cocoons (*CC*), eggs per cocoon (*EPC*), daily mean eggs (*DME*) and cumulative eggs (*CE*), as a dependent variable *Y* on the rearing time and its different powers as more independent ones *Ts*.

Temp. (°C)	Dependent Variable and Derivative as Shown	*n*	*r* or *R*	Regression Equation with *F*-Test Ratio of Each Partial Regression Coefficient	*s_y/x_*	Chi-Square-Test	
						*χ* ^2^	*p*≈
18	Daily mean cocoons, *DMC*	92	−0.8195 ***	*DMC* = 0.816 − (3.74 × 10^−3^)*t* *F_b_*_1_ = 184.0 ***	0.14	4.23	1.0000
21	Daily mean cocoons, *DMC*	92	−0.7546 ***	*DMC* = 1.02 − (4.39 × 10^−3^)*t* *F_b_*_1_ = 119.1 ***	0.21	6.28	1.0000
18	Cumulative cocoons, *CC*	92	0.9986 ***	*CC* = 0.166 + 0.801*t* − (1.88 × 10^−3^)*t*^2^ *F_b_*_1_= 5982.1 ***; *F_b_*_2_= 1255.5 ***	1.30	4.82	1.0000
	Derivative of the equation listed above			*dy/dt* = 0.801 − (3.77 × 10^−3^)*t*			
21	Cumulative cocoons, *CC*	92	0.9984 ***	*CC* = 2.55 + 0.923*t* − (1.81 × 10^−3^)*t*^2^ *F_b_*_1_= 4111.9 ***; *F_b_*_2_= 596.9 ***	1.80	10.7	1.0000
	Derivative of the equation listed above			*dy/dt* = 0.923 − (3.61 × 10^−3^)*t*			
18–21	Eggs per cocoon, *EPC*	52	0.7676 ***	*EPC* = −6.24 + 6.77*t*^1/2^ − 0.878*t* + (4.20 × 10^−3^)*t*^2^ − (1.01 × 10^−5^)*t*^3^ *F_b_*_1_ = 30.6 ***; *F_b_*_2_= 26.6 ***; *F_b_*_3_ = 18.6 ***; *F_b_*_4_ = 14.5 ***	1.12	9.19	1.0000
18	Daily mean eggs, *DME*	92	0.8628 ***	*DME* = −6.78 + 6.59*t*^1/2^ − 0.889*t* + (4.12 × 10^−3^)*t*^2^ − (9.39 × 10^−6^)*t*^3^ *F_b_*_1_ = 39.9 ***; *F_b_*_2_ = 40.4 ***; *F_b_*_3_ = 30.1 ***; *F_b_*_4_ = 23.1 ***	1.07	29.8	1.0000
21	Daily mean eggs, *DME*	92	0.8194 ***	*DME* = −8.78 *+* 9.01*t*^1/2^ *−* 1.28*t +* (6.58 × 10^−3^)*t*^2^ *−* (1.62 × 10^−5^)*t*^3^ *F_b_*_1_ = 33.9 ***; *F_b_*_2_ = 38.2 ***; *F_b_*_3_ = 34.8 ***; *F_b_*_4_ = 31.4 ***	1.59	52.8	0.9994
18	Cumulative eggs, *CE*	92	0.9991 ***	*CE* = −34.4 + 7.83*t* - 0.042*t*^2^ + (9.15 × 10^−5^)*t*^3^ *F_b_*_1_= 2996.1 ***; *F_b_*_2_ = 570.8 ***; *F_b_*_3_ = 222.3 ***	6.94	34.9	1.0000
	Derivative of the equation listed above			*dy/dt* = 7.83 − 0.084*t* + (2.74 × 10^−4^)*t*^2^			
21	Cumulative eggs, *CE*	92	0.9978 ***	*CE* = −23.7 + 9.08*t* − 0.045*t*^2^ + (9.75 × 10^−5^)*t*^3^ *F_b_*_1_ = 1031.7 ***; *F_b_*_2_ = 167.2 ***; *F_b_*_3_ = 64.5 ***	13.7	48.8	0.9999
	Derivative of the equation listed above			*dy/dt* = 9.08 − 0.090*t* + (2.93 × 10^−4^)*t*^2^			

***, significant at *p* < 0.001.

**Table 3 biology-14-00167-t003:** Generational cumulative eggs (*GCE*), generational adult equivalents (*GAE*), and average development rates (*ADR*) in each substage (=a life cycle) of the filial eggs laid by an adult of *Enchytraeus buchholzi* reared at 18 or 21 °C, respectively.

Substage (=a Life Cycle)	18 °C			21 °C		
	*GCE*	*GAE*	*ADR*	*GCE*	*GAE*	*ADR*
1st	101.0	42.5	0.42	101.8	41.2	0.40
2nd	100.9	41.3	0.41	95.6	43.5	0.46
3rd	85.5	48.5	0.57	122.5	66.0	0.54
4th	68.0	32.6	0.48	49.8	29.0	0.58
5th	42.1	23.5	0.56	57.3	29.9	0.52
6th	35.5	21.4	0.60	41.6	19.9	0.48
7th	27.8	11.5	0.41	56.0	31.6	0.56
8th	44.1	23.3	0.53	23.6	13.7	0.58
9th	23.6	13.6	0.58	56.3	25.2	0.45
10th	15.7	9.8	0.62	53.7	29.7	0.55
11th	-	-	-	23.6	14.2	0.60
12th	-	-	-	21.3	10.8	0.51
13th	-	-	-	10.6	5.4	0.51
*m* ± *SE*	54.4 ± 10.1	26.8 ± 4.4	0.52 ± 0.03	54.9 ± 9.4	27.7 ± 4.5	0.52 ± 0.02

**Table 4 biology-14-00167-t004:** Results of the mathematical simulations based on the *AE* of the laboratory population reproduced by an adult of *Enchytraeus buchholzi* reared at 21 °C for a time longer than two generations (*Y*, adult equivalent, *AE*; *T*, date code).

Function	*n*	Linearized Phase		Chi-Square Test		Real Equation	Chi-Square Test	
		*r*	Regression Equation	*χ* ^2^	*p*≈		*χ* ^2^	*p*≈
Exponential	8	0.9812 ***	*y’* = 0.3872 + 0.2181*t*	0.8322	0.9971	*y* = 1.473 × 1.244*^t^*	417.3	0.0000
Logistic	8	−0.9982 ***	*y’* = 6.5606 − 0.2738*t*	0.1281	1.0000	*y* = 709.3/(1 + 706.7*e*^−0.2738*t*^)	6.519	0.4806

***, significant at *p* < 0.001. *y’*, natural log-transformed value of *AE*.

## Data Availability

Most data analyzed and modeled during this study are included in this published article. The rest are available from the corresponding author on reasonable request.

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
