# Peer review of "Reproductive Potential and Population Growth of the Worm Enchytraeus buchholzi (Clitellata: Enchytraeidae) Under Laboratory Conditions as Well as Regression Models"

_biology, 2025, doi:10.3390/biology14020167_

Round 1
Reviewer 1 Report (Previous Reviewer 2)
Comments and Suggestions for Authors
In figure 1, the font size of the ruler should be consistent.
Author Response
Response to Reviewer 1 Comments
Manuscript ID: biology-3412331
|
1. Summary |
|
|
|
Thank you very much for taking the time to review this manuscript. Please find the detailed responses below and the corresponding revisions/corrections highlighted/in track changes in the re-submitted files. I have added the section Simple Summary into our revised manuscript to respond to the journal and/or editors’ requirement. |
||
|
2. Questions for General Evaluation |
Reviewer’s Evaluation |
Response and Revisions |
|
Does the introduction provide sufficient background and include all relevant references? |
Yes |
Thank you for your precious evaluation. |
|
Is the research design appropriate? |
Yes |
Thank you for your precious evaluation. |
|
Are the methods adequately described? |
Yes |
Thank you for your precious evaluation. |
|
Are the results clearly presented? |
Yes |
Thank you for your precious evaluation. |
|
Are the conclusions supported by the results? |
Yes |
Thank you for your precious evaluation. |
3. Point-by-point response to Comments and Suggestions for Authors
Comment 1: In figure 1, the font size of the ruler should be consistent.
Response 1: Thank you for your suggestion! By mentioning “the font size of the ruler”, you may mean the “scale bar” of an image. Figures 1A and 1B were taken with a less-automated microscopic camera in different time; they were not all labeled with a correct scale bar, which was added later after an actual measurement with both ocular and objective micrometers under a microscope. Some areas of images needed magnifying in different rate to illustrate details. As a result, the scale bar changed with magnifications, which led to inconsistent and was unavoidable. In fact, it is something important to calculate size of a target based on scale bar, and its size is reliable if the scale bar is correct.
4. Response to Comments on the Quality of English Language
Point 1: Reviewer 1 checked the entry, “The quality of English does not limit my understanding of the research.”
Response 1: Thank you for checking the entry, which means that the Quality of English Language in the revised manuscript is acceptable.
5. Additional clarifications
[Here, mention any other clarifications you would like to provide to the journal editor/reviewer.]
Response: No, I do not have other clarifications.
Yours sincerely
Limin ZHAO
Corresponding author of “biology-3412331”
23 Jan. 2025
Email: lm_zhao@hotmail.com

Reviewer 2 Report (Previous Reviewer 3)
Comments and Suggestions for Authors
Authors have incorporated the suggestions provided, article can published now.
Author Response
Response to Reviewer 2 Comments
Manuscript ID: biology-3412331
|
1. Summary |
|
|
|
Thank you very much for taking the time to review this manuscript. Please find the detailed responses below and the corresponding revisions/corrections highlighted/in track changes in the re-submitted files. I have added the section Simple Summary into our revised manuscript to respond to the journal and/or editors’ requirement. |
||
|
2. Questions for General Evaluation |
Reviewer’s Evaluation |
Response and Revisions |
|
Does the introduction provide sufficient background and include all relevant references? |
Yes |
Thank you for your precious evaluation. |
|
Is the research design appropriate? |
Yes |
Thank you for your precious evaluation. |
|
Are the methods adequately described? |
Yes |
Thank you for your precious evaluation. |
|
Are the results clearly presented? |
Yes |
Thank you for your precious evaluation. |
|
Are the conclusions supported by the results? |
Yes |
Thank you for your precious evaluation. |
3. Point-by-point response to Comments and Suggestions for Authors
Comment 1: Authors have incorporated the suggestions provided, article can be published now.
Response 1: Thank Reviewer 2 very much for the affirmation! This progress is achieved with all Reviewers’ careful scrutiny and warm support!
4. Response to Comments on the Quality of English Language
Point 1: Reviewer 2 checked the entry, “The quality of English does not limit my understanding of the research.”
Response 1: Thank you for checking the entry, which means that the Quality of English Language in the revised manuscript is acceptable.
5. Additional clarifications
[Here, mention any other clarifications you would like to provide to the journal editor/reviewer.]
Response: No, I do not have other clarifications.
Yours sincerely
Limin ZHAO
Corresponding author of “biology-3412331”
23 Jan. 2025
Email: lm_zhao@hotmail.com

Reviewer 3 Report (New Reviewer)
Comments and Suggestions for Authors
The manuscript is interisting to the research field. Some considerations with the aim of helping to improve the article.
- Format author names - upper and lower cases.
- Abstract is very long. The Abstract need have at most 200 words. In the manuscript, the Abstract have more than 400 words. Then it is necessary reduce it.
- Would be interesting add in the Introduction a paragraph about the importance of control the worms population. What environmental impacts would occur if ther is no control? Perhaps add an explanation mor detailed about the bionomics used in Conclusions.
- Format References in accordance the MDPI rules: Abbreviation Journal names and format year in bold style.
- Add references to:
Equation to calculate Generational Adult Equivalents (GAE) (lines 310-312);
"An actual population possesses two attributes in an ecological niche: temporal and spatial, meaning that it exists and develops in a definite time and space." (lines 694-695).
Author Response
Response to Reviewer 3 Comments
Manuscript ID: biology-3412331
|
1. Summary |
|
|
|
Thank you very much for taking the time to review this manuscript. Please find the detailed responses below and the corresponding revisions/corrections highlighted/in track changes in the re-submitted files. I have added the section Simple Summary into our revised manuscript to respond to the journal and/or editors’ requirement. |
||
|
2. Questions for General Evaluation |
Reviewer’s Evaluation |
Response and Revisions |
|
Does the introduction provide sufficient background and include all relevant references? |
Can be improved |
Thank you! I have increased 10 articles to broaden scientific perspectives of both authors and readers. Please see the text. |
|
Is the research design appropriate? |
Yes |
Thank you for your precious evaluation. |
|
Are the methods adequately described? |
Yes |
Thank you for your precious evaluation. |
|
Are the results clearly presented? |
Yes |
Thank you for your precious evaluation. |
|
Are the conclusions supported by the results? |
Yes |
Thank you for your precious evaluation. |
3. Point-by-point response to Comments and Suggestions for Authors
Reviewer 3 wrote: “The manuscript is interesting to the research field. Some considerations with the aim of helping to improve the article.
Comment 1: Format author names - upper and lower cases.
Response 1: Thank you for your suggestion! Yes, I agree to change the format of author names into “Limin Zhao*, and Guilan Ma – See page 1, line 6.
Comment 2: Abstract is very long. The Abstract need have at most 200 words. In the manuscript, the Abstract have more than 400 words. Then it is necessary reduce it.
Response 2: Thank you for your comment! I ever talked the length of the abstract that contains about 400 words with the Biology Editorial Office (BEO) in May 2024 when the first version of our manuscript was under review. The BEO might have understood our situations because the manuscript comprised two experiments, whose first part contained five dependent variables and the second did two; several regression models were constructed and corresponding relations should be explained clearly… In one word, the experimental study was something very complicated and needed longer abstract and more pages to present! Representing the BEO, one of the editors asked me to add a Simple Summary section before the abstract, no more than 200 words, without refusing the abstract containing about 400 words, which, therefore, is still alive right now. Thank the Biology Editorial Office very much for this kind of arrangement!
Comment 3: Would be interesting add in the Introduction a paragraph about the importance of control the worms population. What environmental impacts would occur if the worm is no control? Perhaps add an explanation more detailed about the bionomics used in Conclusions.
Response 3: Thank you for your comment! Prior to this experimental study, my research team has conducted several studies in other aspects, such as pesticide screen [8], chemical control [9], physiological time [10], field temperature properties [11], bionomics [12], economic benefits and protective measures [6], feeding amount and generational fertility [13]. The worm was re-identified into E. buchholzi in 2022, and thus recovered its original identity [4] (please see page 2, lines 84-88 for detail; numbers in square brackets are numbers of references).
If readers and audience are really interested, please download the references to read, each of which dealt, more or less, with the damage caused by underground pests, including the worm Enchytraeus buchholzi. Even in Figure 2A contained in page 3 of the manuscript, readers can see the terrible losses of seedlings in pesticide-free control plot in an experimental field. Open different locks with different keys. The manuscript is long enough not for me to add some other contents inside. Sorry for that!
Comment 4: Format References in accordance the MDPI rules: Abbreviation Journal names and format year in bold style.
Response 4: Thank you for your suggestion! Yes, as you suggested, I have abbreviated most journal names and written year numbers in bold style. See pages 22-24, lines 878-994.
Comment 5: Add references to: 1) Equation to calculate Generational Adult Equivalents (GAE) (lines 310-312);
Response 5: Thank you for your suggestion! Based on mathematical knowledge and concrete situations, I constructed the equation. Sorry, there is no reference to add.
Comment 6: Add references to: 2) “An actual population possesses two attributes in an ecological niche: temporal and spatial, meaning that it exists and develops in a definite time and space (lines 694-695).”
Response 6: Thank you for your suggestion! The nature has two attributes, temporal and spatial. As part of the nature, all living things, including animals, plants, microorganisms, etc. live in this world, thus they occupy a definite space (an environment with resources) to grow, develop and reproduce with time extension. This is a point of view in philosophy, and my teacher taught me when I was a middle school pupil. This is also a common knowledge (It is not necessary to add references if a knowledge becomes a common one), and I write the two attributes here to explain the situations a special worm population may meet with. There is no concrete, citable reference.
4. Response to Comments on the Quality of English Language
Point 1: Reviewer 3 checked the entry, “The quality of English does not limit my understanding of the research.”
Response 1: Thank you for checking the entry, which means that the Quality of English Language in the revised manuscript is acceptable.
5. Additional clarifications
[Here, mention any other clarifications you would like to provide to the journal editor/reviewer.]
Response: No, I do not have other clarifications.
Yours sincerely
Limin ZHAO
Corresponding author of “biology-3412331”
23 Jan. 2025
Email: lm_zhao@hotmail.com

This manuscript is a resubmission of an earlier submission. The following is a list of the peer review reports and author responses from that submission.
Round 1
Reviewer 1 Report
Comments and Suggestions for Authors
The study is interesting but need to revise a lot
1. Title needs to change accordingly
2. Abstract need to modify (See corrected pdf)
3. Introduction should be provided elaborate data on the work with recent citations and remove old citations
4. Materials method need some corrections (See corrected pdf)
5. Discussion needs significant revision. I did not see relevant citation with comparison.
6. Reference section needs drastic change. I have provided three example citation; but I don't have time to change completely take it an account, the authors shall include many citations while replacing older one

Moderate revision is required for the language part as I've provided in the PDF file
Reviewer 2 Report
Comments and Suggestions for Authors
Can you provide some pictures showed the morphology of the worm and its harmful symptoms?
What is the basis for selecting 18 and 21°C?
Reviewer 3 Report
Comments and Suggestions for Authors
Zhao et al.’s study explores the reproductive potential and population growth of Enchytraeus buchholzi under laboratory conditions, as well as the effects of pruning on plant growth and transcriptome profiles in different tea varieties. This research offers valuable insights and opens new avenues for research in related fields. However, despite its interesting findings, I found numerous inaccuracies in the current article. The text also requires thorough revision by an English expert who understands the subject matter. While I acknowledge the manuscript for publication, it requires revisions before final acceptance.

Zhao et al.’s study explores the reproductive potential and population growth of Enchytraeus buchholzi under laboratory conditions, as well as the effects of pruning on plant growth and transcriptome profiles in different tea varieties. This research offers valuable insights and opens new avenues for research in related fields. However, despite its interesting findings, I found numerous inaccuracies in the current article. The text also requires thorough revision by an English expert who understands the subject matter. While I acknowledge the manuscript for publication, it requires revisions before final acceptance.
Round 2
Reviewer 1 Report
Comments and Suggestions for Authors
I clearly understand that the authors are don't know about the presentation and current research and totally they refused to follow the comments. I am afraid that I cannot review any more.
Comments on the Quality of English LanguageMust be checked by the Native speaker